# Effect of Different Mo Addition on Microstructure and Mechanical Properties of Cu-15Ni-8Sn Alloy

**DOI:** 10.3390/ma15165521

**Published:** 2022-08-11

**Authors:** Kai Zhang, Limin Zhang, Chenglong Shu, Wenxin Fan, Sha Li, Xia Yuan, Junsheng Zhao, Yushuai Wang, Pengfei Wang

**Affiliations:** 1School of Mechanical Engineering, North University of China, Taiyuan 030051, China; 2North Engine Research Institute, Tianjin 300400, China; 3China North Vehicle Research Institute, Beijing 100072, China; 4School of Mechanical and Electrical Engineering, North University of China, Taiyuan 030051, China

**Keywords:** Cu–15Ni–8Sn, discontinuous precipitation, Mo addition, mechanical properties

## Abstract

In the present study, Mo was added to Cu–15Ni–8Sn alloy as the fourth element to solve the limitation of service performance of the alloy by composition design. The phase composition, microstructure transformation and mechanical properties of Cu–15Ni–8Sn–xMo (x = 0.3, 0.9, 1.5 wt.%) alloy were systematically studied by simulation calculation and experimental characterization. The results show that the addition of Mo can improve the as-cast structure of Cu–15Ni–8Sn alloy and reduce segregation and Cu–Mo phase precipitates on the surface with the increase in Mo contents. During solution treatment, Mo can partially dissolve into the matrix, which may be the key to improving the properties of the alloy. Furthermore, the discontinuous precipitation of Sn can be effectively inhibited by adding the appropriate amount of Mo to Cu–15Ni–8Sn alloy, and the hardness of alloy does not decrease greatly after a long-time aging treatment. When Mo content is 0.9 wt.%, the alloy reaches the peak hardness of 384 HV at 4 h of aging. These results provide new ideas for composition optimization of Cu–15Ni–8Sn alloy.

## 1. Introduction

As a typical spinodal hardening copper alloy, Cu–Ni–Sn alloy is widely used in aerospace, aviation, shipbuilding, machinery and other fields due to its high strength, high elasticity, excellent corrosion resistance and stress relaxation resistance [1,2,3,4,5]. It can be used to manufacture bearings, bushings, springs, electronic connectors, etc., and is one of the most prominent substitutes to toxic Cu–Be alloys. Cu–Ni–Sn tends to undergo spinodal decomposition transformation at the early stage of aging (below 520 °C) [6,7]. At the beginning of aging treatment, the DO_22_ and L_12_ ordered phases are transformed from the Sn-rich zone, and the alloy has the highest strength at this stage [8,9,10]. As the aging treatment proceeds, discontinuous precipitation (DP) structure nucleates at the grain boundary and grows rapidly ingrain, which adversely affects the mechanical properties of the alloy [11,12]. This phenomenon is especially serious in Cu–15Ni–8Sn alloy with high Ni and Sn content. Therefore, it is of great practical significance to look for an effective method for inhibiting the DP of Cu–15Ni–8Sn alloy.

Under the traditional smelting process, alloying is often used to improve the microstructure and properties of Cu–Ni–Sn alloy [13,14,15]. It has been reported that the micro-alloying elements in Cu–Ni–Sn alloy has a positive effect on the inhibition of DP [16,17]. The segregation of micro-alloying elements at grain boundaries not only occupies the nucleation site of DP, but also impedes the migration of grain boundaries [18]. On the one hand, adding some micro-alloying elements (Cr, Si, Al, Ti, Co, Fe and Nb) can inhibit the grain growth during heat treatment, and improve organization and performance [19,20,21,22,23,24]. On the other hand, the second phase particles can also be introduced at grain boundaries to inhibit the formation of DP, so as to maintain mechanical properties of the alloy [25,26]. For example, Guo et al. showed that adding Fe element into Cu–15Ni–8Sn alloy can improve the aging resistance and mechanical properties of the alloy [27]. Li et al. believed that Co was distributed in the matrix and precipitated phase in the form of substituted solute, which suppressed discontinuous precipitation to a certain extent, refined grains and improved high-temperature stability [1]. After a long-time aging treatment, the hardness of the alloy is also affected, showing insufficient hardness or rapid decline. As an effective microalloying element, Mo is often used to improve the mechanical properties of alloys [28,29]. However, the effect of Mo contents on the microstructure and properties of Cu–15Ni–8Sn alloys is rarely studied, and there is no relevant theoretical simulation to guide the addition of elements.

In this study, a series of Cu–15Ni–8Sn–xMo alloys were reasonably designed with Mo as an alloying element, and they were processed by melting, solution and aging. The phase composition, microstructure evolution and mechanical properties of alloys with different Mo contents were characterized and analyzed, and the guiding principles of composition design and heat treatment process were put forward, which provided important reference for the optimization of Cu–15Ni–8Sn alloys.

## 2. Materials and Methods

### 2.1. Calculation Method

In the present study, Cu–15Ni–8Sn alloy, one of the commonly used Cu–based bearing alloys, was used as the research object. The pseudo-binary phase diagrams of Cu-15Ni-xSn (x = 0–8 wt.%) and Cu–15Ni–8Sn–xMo (x = 0–1.5 wt.%) systems were calculated by CALPHAD in Pandat 2020 [30,31]. It was used to predict the microstructure transformation of Cu–15Ni–8Sn–xMo (x = 0.3, 0.9, 1.5 wt.%) alloy during solidification. Under the guidance of the above calculation, Cu–15Ni–8Sn–xMo alloy with optimized composition was prepared.

### 2.2. Material Synthesis

The nominal composition of the alloy is Cu–15Ni–8Sn–xMo (x = 0.3, 0.9, 1.5 wt.%), and the corresponding numbers are alloy Ⅰ, Ⅱ and Ⅲ, as shown in Table 1. The Cu–15Ni–8Sn–xMo (x = 0.3, 0.9, 1.5 wt.%) alloy were prepared using electrolytic copper rods (99.95 wt.%), pure nickel blocks (99.95 wt.%), pure tin balls (99.95 wt.%) and molybdenum sheets (99.95 wt.%) as raw materials, melting in vacuum induction furnace. After holding it at 1400 °C for 20 min, the alloy components were fully mixed and poured into steel molds to prepare fixed-size ingot. The ingot was heat-treated at 900 °C for 4 h and then water-quenched immediately after homogenization and solution treatment. Then the solution samples were kept in a resistance furnace for aging treatments (at 400 °C for 1 h to 8 h). The schematic diagram of the overall study is shown in Figure 1.

The content of Mo in the alloy ingot was determined by inductive coupled plasma (ICP). The microstructure characterizations and chemical composition of Cu–15Ni–8Sn–xMo (x = 0.3, 0.9, 1.5 wt.%) alloy were performed using optical microscope (OM; Olympus BX51), scanning electron microscopy (SEM; Zeiss Supra55) equipped with an energy dispersive spectrometer (EDS). For this paper, the SEM secondary electron imaging technology was used, with a voltage of 15 kV. The X-ray diffractometry (XRD, X’Pert Pro) was performed with a Cu Kα radiation and a 2θ scanning rate of 2° per minute from 20° to 90°. OM and SEM observation samples were processed by sandpaper grinding and mechanical polishing before measurement. Vickers hardness tester (Mh-5l) was used to test the microhardness of samples with a load of 200 g. At least 5 points were tested for each sample, and the average value was taken as the final experimental data.

## 3. Results

### 3.1. Phase Composition and Phase Transformation of Cu–15Ni–8Sn–xMo (x = 0, 1.5) Alloy

Figure 2 shows the pseudo-binary phase diagrams of Cu–15Ni–xSn (x = 0–8 wt.%) and Cu–15Ni–8Sn–xMo (x = 0–1.5 wt.%) systems. The comparison shows that the phase diagram becomes very complicated due to the addition of Mo. During the solidification process, BCC phase appears in the pseudo-binary phase diagram of Cu–15Ni–8Sn–1.5Mo alloy, which may be caused by the addition of Mo. A single region of FCC + BCC appears in the phase diagram at about 900 °C, so it is preferred for the solution temperature. At higher temperatures, a large number of γ phases which are harmful to mechanical properties will be produced, while the ordered DO_19_ phase will start to precipitate at about 400 °C. Since the ordered DO_19_ phase will inhibit the DP of γ phase, the optimal aging temperature is set at 400 °C.

Figure 3 shows the phase fraction and Gibbs free energy when Mo content is 1.5 wt.%. It can be seen from Figure 3a that the Gibbs free energy of DO_19_ ordered phase is much lower than other phases. Based on the minimum energy principle, the ordered phase of DO_19_ is formed first. Figure 3b shows that the ordered phase of DO_19_ begins to precipitate at 450 °C and reaches its maximum value at 400 °C. As is well known, the ordered phase can inhibit the DP of γ phase, which further indicates that the optimal aging temperature is 400 °C [16].

### 3.2. Microstructure Analysis of the As-Cast Alloys

Figure 4 shows the as-cast structure of Cu–15Ni–8Sn–xMo (x = 0.3, 0.9, 1.5 wt.%) alloys. The as-cast alloy is composed of typical dendrite structure, which is mainly composed of three parts: matrix phase, black phase and gray phase wrapped by black phase. When the Mo content is 0.3 wt.%, a greyish-white phase appears on the surface of matrix (Figure 4d). With the continuous increase of Mo content to 1.5 wt.%, the grayish-white phase disappears, and white bony precipitates are clearly observed on its surface (Figure 4f). The black structure mainly exists at grain boundaries due to Sn segregation caused by non-equilibrium solidification of the alloy. As can be seen from Figure 4a–c, the as-cast structure of alloy changes slightly with the increase of Mo content. In conclusion, the addition of Mo in Cu–15Ni–8Sn alloy results in a more uniform solidification structure, increased secondary dendrite wall spacing, and reduced Sn segregation.

In order to further explore the form of the second phase in Cu–15Ni–8Sn–xMo (x = 0.3, 0.9, 1.5 wt.%) alloys, the as-cast samples are observed and analyzed by SEM. As shown in Figure 5, Figure 6 and Figure 7, the segregation of Sn element is further demonstrated by comparing the α-Cu matrix phase of Sn poor with the lamellar structure of Sn rich. When Mo content is 0.3 wt.%, the microstructure of alloy shows a second phase structure, but Mo is not detected in the sample. With the increase of Mo content, the bony precipitates appear in the lamellar tissue. As shown in Table 2, Mo is enriched and its atomic content is up to 99.34 and 99.31 wt.%, respectively. Therefore, when the Mo content is low, it exists in the form of a solid solution. When the Mo content increases and exceeds the solubility, it will precipitate in the form of second phase.

### 3.3. Microstructure Analysis of Solid Solution Alloy

Figure 8 shows the metallographic microstructure evolution of Cu–15Ni–8Sn–xMo (x = 0.3, 0.9, 1.5 wt.%) alloys after solution treatment at 900 °C for 4 h. According to Figure 8a–c, the dendrites completely disappear after solution treatment at 900 °C, and there are two matrix phases with different colors on the surface of Cu–15Ni–8Sn–xMo alloy. With the Mo content increases to 1.5 wt.%, white stripe or a granular structure are precipitated in the dark phase. It is found that with the increase of Mo content, some black tissues grow from the grain boundary to the grain interior more and more intensely, which may be Sn-rich phase. As can be seen from Figure 8d–f, the second phase still exists after solid solution treatment. With the increase of Mo content, the precipitation of second phase can be promoted. This may be because Mo is dissolved in the matrix and occupies part of coordination.

According to the above metallographic analysis, the second phase still exists on the surface after solid solution treatment. In order to determine the existence form of elements in the solid solution state, samples are carried out by SEM and EDS. When Mo content is 0.3 wt.%, precipitates exist in grains and grain boundaries, but Mo is not detected (Figure 9a–c). By observing the alloy with Mo contents of 0.9 and 1.5 wt.% (Figure 9d–e), sharp black bony precipitates can be found on the surface. The results show that 0.3 wt.% Mo can be dissolved into the matrix as solid atoms after solution treatment, but when the content of Mo reaches 0.9 and 1.5 wt.%, it will exist in the form of second phase.

### 3.4. Microstructure Analysis of Alloy after Aging

According to Zhao et al. [7], the TTT diagram of Cu–15Ni–8Sn alloy was summarized. Early spinodal decomposition takes place, and the modulation structure of Sn-poor region and Sn-rich region is formed at aging 400 °C. Subsequently, the ordered phases DO_22_ and L_12_ are precipitated from the Sn-rich region, resulting in a continuous increase of coherent stress with the matrix. But DP is formed at the grain boundary during later aging, which is harmful for mechanical properties of the alloy.

Figure 10 shows the metallographic microstructure of three alloys with different Mo contents at 400 °C for 1–8 h, black phase firstly precipitates from the grain boundary. With the increase in aging time, the DP grows smoothly from grain boundary to matrix. As show in Figure 10a_1_–e_1_, the addition of appropriate Mo significantly inhibits the nucleation and slows down the growth of DP. Furthermore, the amount of black phase precipitation increases with the increase of Mo content in the same aging time. This indicates that excessive Mo contents (0.9, 1.5 wt.%) promote the nucleation and growth of DP during aging treatment.

### 3.5. Microstructure Analysis

Figure 11 shows the XRD patterns of Cu–15Ni–8Sn–xMo (x = 0.3, 0.9, 1.5 wt.%) alloys in as-cast state and after solution treatment at 900 °C for 4 h. It can be seen from Figure 11a that the content of Mo in the as-cast alloy has no effect on the diffraction peak of copper, which further indicates that Mo is not solidly dissolved in the matrix. According to Figure 11b, when Mo contents increase to 0.9 and 1.5 wt.%, strong Cu–Mo/Cu diffraction peaks appear in the solid solution alloy, which is consistent with the previous analysis. In addition, there is no Cu-Mo diffraction peak founding in solid solution alloy when Mo content is 0.3 wt.%, which may be related to the low Mo content.

### 3.6. Hardness

Figure 12 shows the hardness changes of Cu–15Ni–8Sn–xMo (x = 0.3, 0.9, 1.5 wt.%) alloys with aging time from 1 h to 8 h. For the super-saturated solid solution without aging treatment, the hardness is the highest when Mo content is 1.5 wt.%. This is due to the existence of Mo in the matrix as solid solution atoms, which plays a role of solid solution strengthening. According to solution strengthening mechanism, Ⅲ alloy with high concentration of solution atoms has higher hardness. At the early aging stage, the alloy hardness increases rapidly, which is attributed to the amplitude modulation decomposition transformation and the continuous precipitation of ordered phase [32,33,34]. It is well known that the hardness of Cu–15Ni–8Sn alloy will decrease significantly after a long time of aging, which is caused by DP [12]. The DP is inhibited and the hardness of the alloy does not decrease greatly by adding Mo after a long time of aging treatment. However, excess Mo contents may promote the precipitation of DP. For example, the hardness of the Ⅲ alloy decreased slightly, because DP products increase after a long time of aging.

## 4. Conclusions

A series of Cu–15Ni–8Sn–xMo alloys with excellent mechanical properties and high hardness were obtained in the present study. The effect of Mo contents on the phase composition, microstructural evolution and phase transformation of Cu–15Ni–8Sn–xMo alloys was investigated. And the effect of phase composition and transformation on the mechanical properties of the alloy was revealed. The main conclusions are as follows: with the addition of Mo, the segregation of the Sn element in the as-cast structure can be slightly improved, and the structure is uniform. With the increase in Mo contents to 0.9 wt.% and 1.5 wt.%, Mo precipitation is precipitated on the surface of alloy. By solution treatment, the Mo–rich phase cannot be completely dissolved into the matrix. In the aging process, an appropriate amount of Mo can significantly inhibit the nucleation and growth of DP, while excessive Mo can promote it. The hardness of Cu–15Ni–8Sn–xMo (x = 0.3, 0.9, 1.5 wt.%) alloy reaches its peak after aging at 400 °C for 4 h, and there is no significant decrease in hardness at the later stage of aging.

## Figures and Tables

**Figure 1 materials-15-05521-f001:**
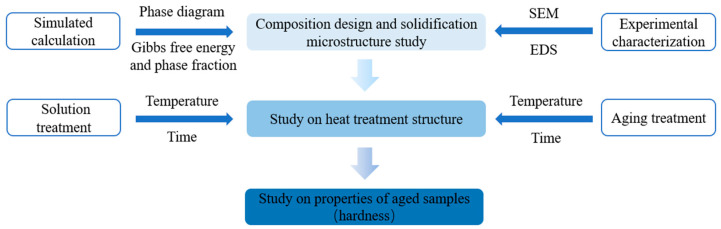
Schematic diagram of the overall research scheme.

**Figure 2 materials-15-05521-f002:**
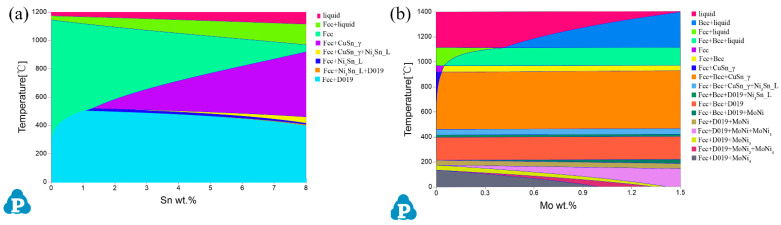
The simulation-calculated pseudo-binary phase diagrams of (**a**) Cu–15Ni–xSn (x = 0–8 wt.%) and (**b**) Cu–15Ni–8Sn–xMo (x = 0–1.5 wt.%) systems.

**Figure 3 materials-15-05521-f003:**
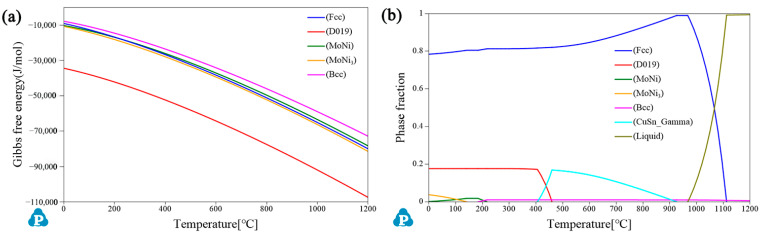
(**a**) Gibbs free energy diagram and (**b**) Phase fraction diagram at 1.5 wt.% Mo addition.

**Figure 4 materials-15-05521-f004:**
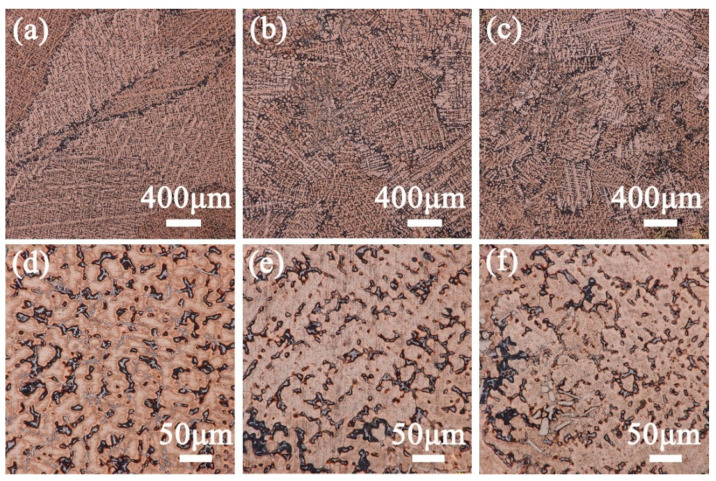
Metallurgical structure of as-cast Cu–15Ni–8Sn–xMo alloys with different Mo content: (**a**,**d**) 0.3, (**b**,**e**) 0.9, (**c**,**f**) 1.5 wt.%.

**Figure 5 materials-15-05521-f005:**
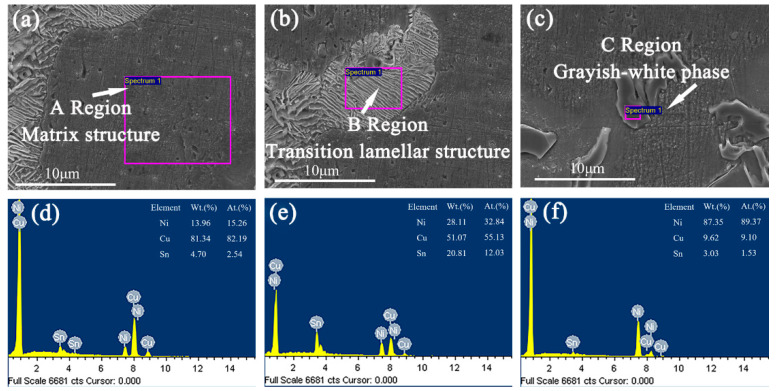
The SEM images of as-cast structure for Cu–15Ni–8Sn–0.3Mo alloy: (**a**,**d**) matrix structure, (**b**,**e**) transition lamellar structure, (**c**,**f**) grayish-white phase.

**Figure 6 materials-15-05521-f006:**
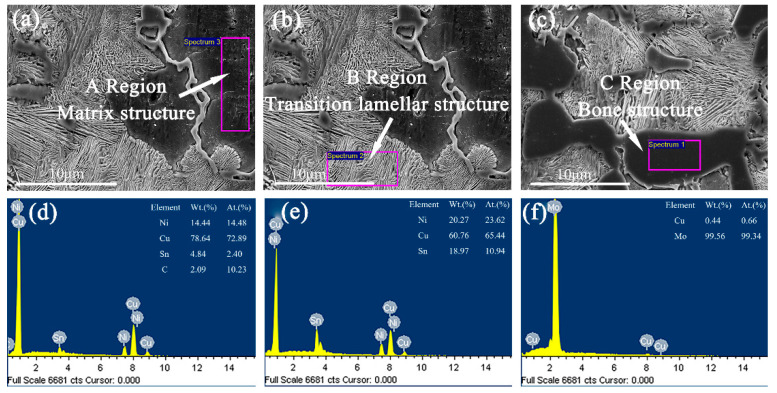
The SEM images of as-cast structure for Cu–15Ni–8Sn–0.9Mo alloy: (**a**,**d**) matrix structure, (**b**,**e**) transition lamellar structure, (**c**,**f**) bone structure.

**Figure 7 materials-15-05521-f007:**
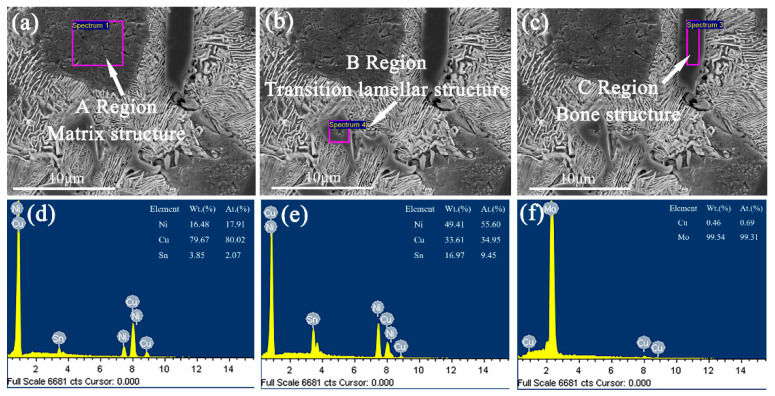
The SEM images of as-cast structure for Cu–15Ni–8Sn–1.5Mo alloy: (**a**,**d**) matrix structure, (**b**,**e**) transition lamellar structure, (**c**,**f**) bone structure.

**Figure 8 materials-15-05521-f008:**
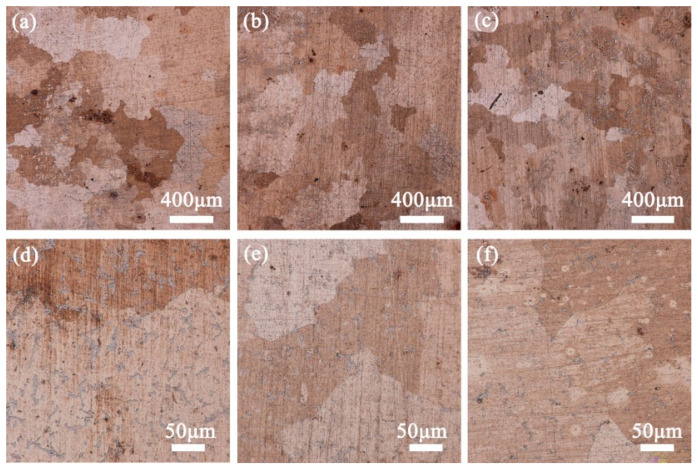
Metallurgical structure of Cu–15Ni–8Sn–xMo alloy after solution at 900 °C for 4 h: (**a**,**d**) x = 0.3, (**b**,**e**) x = 0.9, (**c**,**f**) x = 1.5 wt.%.

**Figure 9 materials-15-05521-f009:**
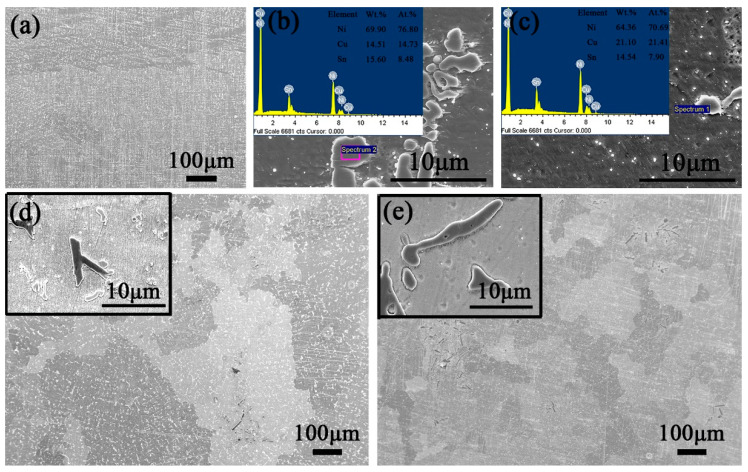
The SEM images of solid solution microstructure for Cu–15Ni–8Sn–xMo alloys with different Mo content: (**a**–**c**) x = 0.3, (**d**) x = 0.9, (**e**) x = 1.5 wt.%.

**Figure 10 materials-15-05521-f010:**
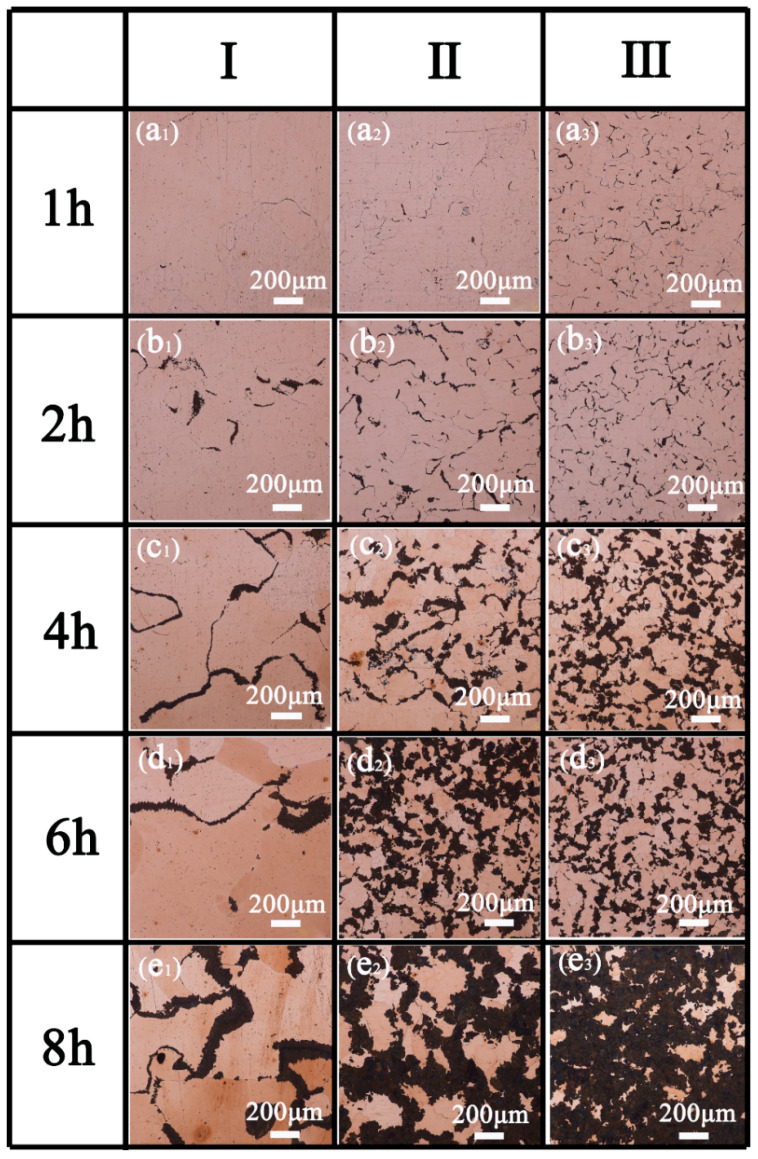
Metallographic microstructure of Cu–15Ni–8Sn–xMo (x = 0.3, 0.9, 1.5 wt.%) alloys aged at 400 °C for 1–8 h: (**a_1_**–**e_1_**) x = 0.3, (**a_2_**–**e_2_**) x = 0.9, (**a_3_**–**e_3_**) x = 1.5 wt.%.

**Figure 11 materials-15-05521-f011:**
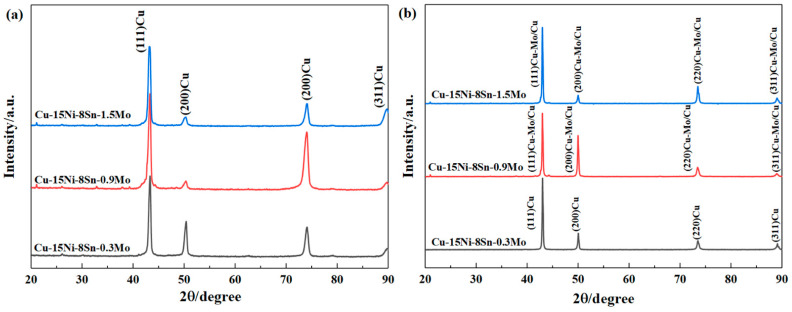
(**a**) XRD patterns of Cu–15Ni–8Sn–xMo (x = 0.3, 0.9, 1.5 wt.%) alloy at as-cast state and (**b**) after solid solution treatment at 900 °C for 4 h.

**Figure 12 materials-15-05521-f012:**
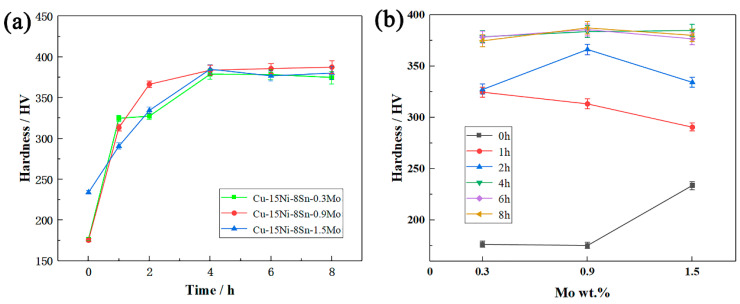
Aging hardening curve of Cu–15Ni–8Sn–xMo (x = 0.3, 0.9, 1.5 wt.%) alloy at 400 °C isothermal aging: (**a**) time–hardness diagram, (**b**) Mo content–hardness diagram.

**Table 1 materials-15-05521-t001:** Chemical compositions of the experimental alloys.

Alloys	I	II	III
Mo nominal/measured composition (wt.%)	0.3/0.256	0.9/0.825	1.5/1.44

**Table 2 materials-15-05521-t002:** Atomic weight percentage of Cu–15Ni–8Sn–xMo (x = 0.3, 0.9, 1.5 wt.%) alloys.

Sample		Ni (at. %)	Cu (at. %)	Sn (at. %)	Mo (at. %)
I	A	15.26	82.19	2.54	–
	B	32.84	55.13	12.03	–
	C	89.37	9.10	1.53	–
II	A	14.48	72.89	2.40	–
	B	23.62	65.44	10.94	–
	C	–	0.66	–	99.34
III	A	17.91	80.02	2.07	–
	B	55.60	34.95	9.45	–
	C	–	0.69	–	99.31

## Data Availability

Not applicable.

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
