# Peer review of "Effect of Different Mo Addition on Microstructure and Mechanical Properties of Cu-15Ni-8Sn Alloy"

_materials, 2022, doi:10.3390/ma15165521_

Round 1

Reviewer 1 Report

In review of the manuscript, titled: “Effect of Different Mo addition on microstructure and mechanical properties of Cu-15Ni-8 Sn alloy”, the manuscript is written and designed very well. But there exist some minor flaws which must be addressed before publication.

1.       What is the significance of selecting the specific material? Please add it in introduction portion.

2.       Problem statement is missing to carry this work, add few lines in the introduction portion.

3.       For material synthesis, please clarify either Cu-15Ni-8Sn is prepared first and then Mo is added or the Cu-15Ni-8Sn-xMo (x=0.3, 0.9, 1.5 wt%), is fabricated directly?

4.       In Figure 3 (a-f), authors claim the figures belong to Cu-15Ni-8Sn-xMo (x=0.3, 0.9, 1.5) samples. But it is never clear in the images and not even in the literature which Figure belongs to which sample? Please add the specific formulae in the specific Figure

5.       The figures 3-6 must be compared with the pure Cu-15Ni-8Sn alloy for clear depiction.

6.       If the authors can provide the colored mapping for the SEM images, it will be more beneficial for the readers to understand the phenomenon.

7.       Same issues are with the Figure 7 and 8 as described in point 4. Please mention the concentration of the material in the images.

8.       Even with the XRD analysis pure Cu-15Ni-8Sn without Mo sample is ignored for comparison.

9.       Authors claim” With the increase of Mo content, Cu-Mo phase is precipitated”, provide its evidence.

10.   In section 3.6, develop a clear correlation for hardness variations with microstructural changes (Hardness variations should be associated with microstructural phenomenon). Moreover, there is no support provided to analysis by the relevant references. The authors are recommended to add recent references for example.      1. https://doi.org/10.1016/j.ceramint.2022.04.115,  2.  https://doi.org/10.3390/met8080620,3. https://doi.org/10.3390/cryst11091115

Reviewer 2 Report

1.      Line 6-13, if the email of the authors in one place is only one, it does not need to give the initial author's name after the author's email. Revise it.

2.      Quantitative results need to be added rather than only qualitative in the abstract section.

3.      Reorder keywords based on alphabetical order.

4.      What is the novelty of the present work? Different Mo addition studies in metal alloys have been widely studied in the past. Nothing really new or significant contribution to the present research. It is the lack of novel and contribution that leads this manuscript should be rejected. The authors should aware of these points.

5.      Specific one-by-one previous studies with their findings and their shortcomings are needed to be explained in the introduction section.

6.      Materials alloy is impacting their performance, for example in the medical implant. The authors need to explain this important point in the introduction and/or discussion section. Also, to support this explanation, a suggested reference published by MDPI is needed to be adopted as follows: Computational Contact Pressure Prediction of CoCrMo, SS 316L, and Ti6Al4V Femoral Head against UHMWPE Acetabular Cup under Gait Cycle. J. Funct. Biomater. 2022, 13, 64. https://doi.org/10.3390/jfb13020064

7.      Section 2 name is needed to change to “Materials and Methods”.

8.      The authors need to give additional illustrative figures to present the workflow of the studies to make sure the reader is easier to understand and more interested rather than only text with specific figures.

9.      Standard/basis/protocol is every procedure is needed more given information.

10.   More information regarding tools specification is needed to be stated.

11.   Accuracy and tolerance of the tools used are impacting to the results that should be explained, every tool should be mentioned their accuracy and tolerance to make the reader understand different results in the further study.

12.   Results comparison with similar/Identical studies in the past is needed in the discussion section.

13.   The limitations of the present study should be explained.

14.   The conclusion is needed to be arranged into paragraphs, not point by point as present form.

15.   Further study is needed to explain in the conclusion section.

16.   English used needs to be proofread due to grammar and language style issues.

17.   Please, make sure the authors have used the Materials, MDPI format properly. Authors can download the published version to compare it with the authors’ document.

Reviewer 3 Report

Zhang et al. have investigated the effect of Mo addition to Cu-15Ni-8Sn alloy. The Mo addition to this alloy is relatively new topic; the choice of topic is sufficiently proved by the compact introduction. Strength of the paper is, that the selection of the Mo content investigated was based on theoretical calculations / simulations. However, the paper has serious problems (eg. contradictory conclusions), therefore I cannot recommend you to publish it in Materials in present form.

General issues:

    The English of the article is very poor, especially regarding to singular and plural and tenses. The use of "Mo", "Sn", "Mo element", "Sn element" etc. should be consequent.
    The figure captions lack of information and do not help to identify the contents of the subfigures. On Fig. 2 (a) and (b) the corresponding phases should be marked with the same color. Gibbs free energy has an unit.

Serious problems:

    Lines 114-115: authors suggest, that the microstructure of the as-cast alloys was changed significantly by the Mo addition.
        while
    Lines 125-126: authors suggest, that the microstructure of the as-cast alloys was not changed significantly by the Mo addition.

    Lines 130-133: authors suggest, that in case of alloy III, Mo was precipitated as a second phase.
        while
    Lines 197-199: authors suggest, that in case of alloy III, Mo was in the matrix as solid solution form.

Major questions:

    Section 3.5.: if Mo is not in solid solution with Cu/Ni/Sn, where are the Mo diffraction peaks? How was the Cu and Cu-Mo diffraction peaks distinguished? Fig 10 (b): why the Cu-Mo (200) diffraction peak is significantly smaller in case of 1.5% Mo than case of 0.9%? Lines 185-186: if Mo has no effect on the Cu diffraction peaks, why are the three diffraction patterns on fig. 10 (a) significantly different? Fig. 10.: what are the other peaks on the figure?

    Lines 128-130: if no Mo was detected in alloy I, how the result of table 1. (0.256 wt%) was created?

    Section 3.2.: captions of figures 4-6 should mention should mention that which subfigure corresponds to which morphologies (matrix, lamellar, bone). On Fig 5. what is the difference between the selected areas of (a) an (c) which can be observed by SEM? (What was the reason of the selection of these areas? I cannot see any significant difference between them (except the EDS spectra).) Fig 5. (a): please discuss about the significant amount of carbon.  On Fig. 6 (b) why not a lamellar area was selected?

Minor questions:

    Section 3.3.: Figure 8.: which is that sample? An other SEM/EDS pair also should be presented where the Mo content could be observed.

Major issues:

    Lines 114-115: authors suggest, that the significant microstructural changes could be seen on Fig. 3. I cannot see any relevant differences between Fig. 3 (b)-(d)-(f) or (c)-(e).

    Section 3.2.: EDS results should be summarized in a table.

    Section 3.2.: the three kind of "parts" (line 112: matrix phase, black phase and gray phase) should be connected with the three kind of "morphologies" (line 124: matrix structure, transition lamellar structure and bone structure).

Minor issues:

    Fig. 3.: subfigures should be rearranged as the first line for the low magnification images and the second one for the high magnification images.

    Lines 104 and 203: Phrases "it is well known that..." should be corroborated with references.

    There are missing experimental details: X-ray radiation source (wavelength), SEM energy and signal type, EDS energy.

    Line 64: CALPHAD should be referenced.

Round 2

Reviewer 2 Report

Good job from the Authors. My comments have been addressed.

Author Response

Thank you very much for finding our revision reasonable.

Reviewer 3 Report

General issues:

    It is unnecessary to start every answer with a thank you.

    There are still parallel use of "Mo" and "Mo element" (and others). "Element" is unnecessary.

Serious problems:

    Lines 140-141: authors suggest that the addition of Mo has not significantly changed the microstrucutre. However they presents that Cu-15Ni-8Sn-0.3Mo contains Ni rich precipitations (bone structure), while Cu-15Ni-8Sn-0.6Mo and Cu-15Ni-8Sn-1.5Mo contain (almost) pure Mo precipitations in the bone structure. This is a contradiction.

    It is a very serious problem, that no Mo has been detected with EDS in sample Cu-15Ni-8Sn-0.3Mo neither as a solid solution in the matrix nor as precipitations. Authors claim that is due to uneven distribution (line 145). These results suggest that Alloy-I has a significantly different microstructure than Alloy-II and Alloy-III which must be investigated and discussed more.

    Lines 147-149: authors suggest that in the as cast alloys II and III Mo is in the form as a second phase (precipitations) not in solid solution.
    while
    Lines 214-217: authors suggest that Alloy-III contains Mo in the form of solid solution atoms. This is a contradiction and not supported by the previous results.

    Lines 174-179 and Fig 9.: authors conclude, that due to heat treatment, Mo was dissolved into the matrix as solid solution.
        a) Conclusion is about all three samples, but fig. 9. is only about Cu-15Ni-8Sn-0.3Mo.
        b) SEM/EDS (fig. 9.) does not show any presence of Mo (neither precipitations nor solid solution).
        c) The lack of Mo during the EDS measurement was explained in line 144 by "low Mo content and uneven distribution". Here is explained by solid solution.

Minor issues:

    Lines 71-72 and figure 2.: these are the pseudo binary phase diagrams of Cu-15Ni-xSn (x=0-8) and Cu-15Ni-8Sn-xMo (x=0-1.5) systems.

    Line 237: that Cu-Mo phase would be Mo precipitation.

    In table 2, the two horizontal lines separating "II" should be elongated.

    Line 81.: typo: "20 min"

Round 3

Reviewer 3 Report

It is still embarrassing, that ICP and EDS have showed different results in case of Cu-15Ni-8Sn-0.3Mo. During further researches, similar contradictories like this must be resolved.